**Technical note: Measurement of chemically-resolved volume equivalent diameter and effective density of particles by AAC-SPAMS**

Long Peng[1,2], Lei Li[4], Guohua Zhang[1, 3*], Xubing Du[4], Xinming Wang[1, 3], Ping'an Peng[1, 3], Guoying Sheng[1], Xinhui Bi[1, 3*]

[1] State Key Laboratory of Organic Geochemistry and Guangdong Provincial Key Laboratory of Environmental Protection and Resources Utilization, Guangzhou Institute of Geochemistry, Chinese Academy of Sciences, Guangzhou 510640, China

[2] University of Chinese Academy of Sciences, Beijing, 100049, China

[3] Guangdong-Hong Kong-Macao Joint Laboratory for Environmental Pollution and Control, Guangzhou 510640, China

[4] Institute of Mass Spectrometer and Atmospheric Environment, Jinan University, Guangzhou 510632, China

*Correspondence to: bixh@gig.ac.cn and zhanggh@gig.ac.cn

**Abstract**

Size and effective density ($\rho_e$) are important properties of aerosol particles and are related to their influences on human health and the global climate. The volume equivalent diameter ($D_{ve}$) is an intrinsic property that is used to evaluate particle size. Three definitions of $\rho_e$ are generally used to characterize the physical property of a particle as an alternative to particle density, in which only the $\rho_e^{II}$, defined as the ratio of particle density ($\rho_p$) to a dynamic shape factor ($\chi$), has the characteristic of being independent of particle size. However, it is still challenging to simultaneously characterize the $D_{ve}$ and $\rho_e^{II}$ of aspherical particles. Here, we present a novel system that classifies particles with their aerodynamic diameter ($D_a$) by aerodynamic aerosol classifiers (AAC) and determines their vacuum aerodynamic diameter ($D_{va}$) by single particle aerosol mass spectrometry (SPAMS) to achieve a measurement of $D_{ve}$ and $\rho_e^{II}$. The reliability of the AAC-SPAMS system for accurately obtaining $D_{ve}$ and $\rho_e^{II}$ is verified based on the results that the deviation between the measured and theoretical values is less than 6% for the size-resolved spherical polystyrene latex (PSL). The AAC-SPAMS system is applied to characterize the $D_{ve}$ and $\rho_e$ of $(NH_4)_2SO_4$ and $NaNO_3$ particles, suggesting that these particles are aspherical and their $\rho_e$ are independent of particle size. Finally, the AAC-SPAMS system is deployed in a field measurement, showing that it is a powerful technique to characterize the chemically-resolved $D_{ve}$ and $\rho_e^{II}$ of particles in real time.

## 1. Introduction

Size and particle density ($\rho_p$) are critical parameters of aerosol particles in quantifying the  impact of aerosols on air quality, human health and global climate change (Buseck and Posfai, 1999; Poschl, 2005; Pitz et al., 2003). Effective density ($\rho_e$) has been adopted to characterize the physical property of a particle as an alternative to $\rho_p$, since $\rho_p$ for aspherical aerosol particles is hardly measured (Sumlin et al., 2018; Katrib et al., 2005). Size and $\rho_e$ govern the transport properties of a particle both in the atmosphere and in the human respiratory system (Seinfeld and Pandis, 1998; Liu and Daum, 2008) and directly and/or indirectly influence the potential of the particle to absorb or reflect solar radiation (Tang, 1997; Zhao et al., 2019; Liu and Daum, 2008). $\rho_e$ can also provide information concerning particle morphology (Yon et al., 2015) and serve as a tracer for atmospheric processing (Guo et al., 2014; Yin et al., 2015; Liu et al., 2015). However, the quantitative relationship between aerosol properties, namely, size and $\rho_e$, and their effects on air quality, human health and global climate change is not yet well understood, which is partly because important aerosol properties cannot be measured by current techniques.

**Size.** Size is a fundamental property of particles, which can be parameterized by the physical quantity of volume equivalent diameter ($D_{ve}$). Defined as the diameter of a spherical particle with the same volume as the particle (DeCarlo et al., 2004), $D_{ve}$ is an intrinsic physical quantity that can be used to evaluate the actual size of the particle. However, to date, atmospheric science usually describes particle size by other diameter definitions, such as the electric mobility diameter ($D_m$), aerodynamic equivalent

diameter ($D_a$) and vacuum aerodynamic equivalent diameter ($D_{va}$), whose relationships
with $D_{ve}$ are shown in Eqs. (1)-(3), respectively:

$$\frac{D_m}{C_c(D_m)} = \frac{D_{ve}}{C_c(D_{ve})} \chi_t, \tag{1}$$

$$D_a = D_{ve} \sqrt{\frac{\rho_p C_c(D_{ve})}{\chi_t \cdot \rho_0 \cdot C_c(D_a)}}, \tag{2}$$

$$D_{va} = \frac{\rho_p}{\rho_0} \frac{D_{ve}}{\chi_v}, \tag{3}$$

where $C_c(D)$ is the Cunningham slip correction factor, $\chi_t$ and $\chi_v$ represent the aerosol
dynamic shape factor ($\chi$) in the transition regime and in the free-molecule regime,
respectively, and $\rho_0$ represents the unit density of 1.0 g/cm$^3$. From these definitions, it
can be seen that $D_m$, $D_a$, and $D_{va}$ are originally derived from $D_{ve}$, but in actuality, they
do not reflect the actual size of the aspherical particle. Meanwhile, $D_{ve}$ of aspherical
particles cannot be easily obtained, which limits its application in the scientific
community.
**Effective density.** At present, three definitions of $\rho_e$ are introduced in atmospheric
science (DeCarlo et al., 2004): the first definition ($\rho_e^I$) is the ratio of the measured
particle mass ($m_p$) to the particle volume ($V$) calculated assuming a spherical particle
with a diameter equal to the measured $D_m$; the second definition ($\rho_e^{II}$) is the ratio of $\rho_p$
to $\chi$ (Hand and Kreidenweis, 2002); and the third definition ($\rho_e^{III}$) is the ratio of $D_m$ and
$D_{va}$, all of which are expressed in Eqs. (4)-(6), respectively.

$$\rho_e^I = \frac{6m_p}{\pi D_m^3} \tag{4}$$

$$\rho_e^{II} = \frac{\rho_p}{\chi} \tag{5}$$

$$\rho_e^{III} = \frac{D_{va}}{D_m} \rho_0 \tag{6}$$

The definitions of $\rho_e^I$ and $\rho_e^{III}$ can be derived into the final forms, as shown in the Eqs.(7)
and (8), respectively.

$$\rho_e^{\mathrm{I}} = \frac{\rho}{{\chi_t}^3} \cdot \left(\frac{C_c(D_{ve})}{C_c(D_m)}\right)^3 \tag{7}$$

$$\rho_e^{III} = \rho \cdot \frac{C_c(D_{ve})}{\chi^2 \cdot C_c(D_m)} \tag{8}$$

The Eq. (7) is derived from combining the Eq. (1) with Eq. (4), in which $m_p$ is equal to
$1/6\,\rho \cdot D_{ve}^3$. The detailed derivation of Eq. (8) was presented in Schneider et al. (2006). A
variety of methods are developed to characterize $\rho_e^I$ and $\rho_e^{III}$, among which the more
advanced methods are to achieve the measurement of the chemically-resolved effective
density. Combining a single particle soot photometer (SP2) with a (volatility) tandem
differential mobility analyser ((VT)DMA) can measure the $\rho_e^I$ of particles mixed with
soot (Zhang et al., 2016b; Wu et al., 2019; Han et al., 2019). The measurement of
chemically-resolved $\rho_e^{III}$ can be achieved by coupling a DMA with an on-line aerosol
mass spectrometer such as Single Particle Laser Ablation Time-of-Flight Mass
Spectrometer (SPLAT) (Zelenyuk et al., 2005; Zelenyuk et al., 2006; Alexander et al.,
2016), an aerosol mass spectrometer (AMS) (Dinar et al., 2006; Schneider et al., 2006;
Kiselev et al., 2010), an aerosol time-of-flight mass spectrometer (ATOFMS) (Spencer
and Prather, 2006; Spencer et al., 2007), and single-particle aerosol mass spectrometry
(SPAMS) (Zhang et al., 2016a; Zhai et al., 2017). However, the $\rho_e^I$ and $\rho_e^{III}$ are
demonstrated to have the inherent characteristics of decreasing with increasing particle
size, which will be presented in a separate publication. Therefore, it will introduce
systemic error when assessing the particle impacts on visibility, human health and
climate change from the physical quantities in $\rho_e^I$ and $\rho_e^{III}$. In contrast, $\rho_e^{II}$ is independent
of particle size. For example, for soot particles with $\chi$ of 2.5 and $\rho_p$ of 1.80 g/cm³, the
calculated $\rho_e^I$, $\rho_e^{II}$, and $\rho_e^{III}$ are 0.43, 0.72, and 0.45 g/cm$^3$ at $D_m$ of 40 nm, and 0.22,
0.72, and 0.36 g/cm$^3$ at $D_m$ of 550 nm, respectively. The big gap between the three
definitions of effective density suggests that they should be carefully treated when
characterizing the particles. However, the $\rho_e^{II}$ has not been widely applied in
atmospheric sciences because of the lack of measurement techniques. Previous
literatures tried to retrieve the $\rho_e^{II}$ and the real part in the refractive index ($n$) through a
fitting procedure that compares the measured light-scattering intensity of particles
($R_{meas}$) to the theoretical values ($R_{theory,test}$) calculated by a series of $n$ and $\rho_e^{II}$ values
(Moffet and Prather, 2005; Moffet et al., 2008; Zhang et al., 2016a). Moffet and Prather
(2005) successfully obtained $\rho_e^{II}$ for spherical particles by single particle mass
spectrometry. However, subject to the accuracy of Mie theory for the aspherical
particles, dry NaCl and calcium-rich dust particles were failed to fit the $R_{theory,test}$ well
to $R_{meas}$ (Moffet et al., 2008). Similarly, Zhang et al. (2016a) failed to simultaneously
retrieve $\rho_e^{II}$ and $n$ for $(NH_4)_2SO_4$ and $NaNO_3$ particles. To our best knowledge, there is
no appropriate technique to achieve the measurement of $\rho_e^{II}$ for aspherical particles.
The aim of the present work is to develop a method to simultaneously obtain $D_{ve}$ and
$\rho_e^{II}$ for aspherical particles. For simplicity, the symbol $\rho_e$ in the following text refers to
the definition of $\rho_e^{II}$. The established system of an aerodynamic aerosol classifier
(AAC)-SPAMS is capable of characterizing the $D_a$ and $D_{va}$ of particles, which can be
applied to theoretically derive $D_{ve}$ and $\rho_e$. To verify the reliability of the AAC-SPAMS
system, we apply it to measure the $D_{ve}$ and $\rho_e$ of the spherical particles of polystyrene
latex (PSL). The results are in good agreement with the theoretical values. Finally, the
AAC-SPAMS system is applied to measure the $D_{ve}$ and $\rho_e$ for $(NH_4)_2SO_4$ and $NaNO_3$
particles and for the chemically-resolved atmospheric particles.

**2.  Experimental section**
**2.1 Measurement system**
Figure 1 shows a schematic diagram of the AAC-SPAMS system. The particles are
first dried by a diffusion drying tube (TSI 9302, USA), classified by AAC (Cambustion
Ltd., UK) based on the aerodynamic diameters $D_a$, and then transported into SPAMS in
which the $D_{va}$ and the mass spectra of individual particles are obtained. The working
principle of the AAC is described in detail elsewhere (Tavakoli and Olfert, 2013). AAC
consists of two coaxial cylinders that rotate at the same rotational speed. Polydisperse
particles enter into the space between the cylinders (i.e., classification column) and
experience a centrifugal force that causes them to move toward the outer cylinder. The
particles to be classified can leave the classification column with the particle-free sheath
flow and finally exit the AAC with the sample flow. Thus, the $D_a$ values of classified
particles can be derived from their relationship with their relaxation time ($\tau$), as shown
in Eq. (9):
$$\tau = \frac{C_C(D_a) \cdot \rho_0 \cdot D_a^2}{18\mu} \tag{9}$$
where $\mu$ is the gas dynamic viscosity. Particles with large relaxation times impact and
adhere to the outer cylinder, while particles with small relaxation times exit the
classifier with the exhaust flow. The exhaust flow from the AAC was about 0.3 lpm,
and the Size Resolution Parameter (Rs) of the AAC was set as 40.
Detailed information about the operation of SPAMS (Hexin Analytical Instrument
Co., Ltd., China) is given elsewhere (Li et al., 2011). Briefly, the particles are
introduced into the vacuum system through a 0.1 mm critical orifice and are gradually
collimated into a beam in the aerodynamic lens. Two continuous diode Nd:YAG laser
beams (532 nm) are used to aerodynamically size the particles, which are subsequently
desorbed/ionized by a pulsed laser (266 nm) that is triggered based on the velocity of a
specific particle. The generated positive and negative ions are recorded with the
corresponding particle size. The $D_{va}$ of the particle is related to the transit time between
the two laser beams (532 nm) in SPAMS, which can be obtained by using a calibration
curve generated from the measured transit times of a PSL series with predefined sizes
(nominal diameters).

**2.2 Laboratory experiments**
Dried spherical PSL (Nanosphere Size Standards, Duke Scientific Corp., Palo Alto)
($\rho_p = 1.055$ g/cm$^3$ and $\chi = 1.0$) with $D_{ve}$ values of $203 \pm 5$ nm, $310 \pm 6$ nm, $510 \pm 5$ nm,
and $740 \pm 6$ nm were used in the AAC-SPAMS system, and the $D_{ve}$ was verified by
Scanning Mobility Particles Sizer (Model 3938, TSI Inc., USA). The PSL particles were
first classified by AAC, and then their $D_{va}$ values were obtained by SPAMS. ACC-
SPAMS was also applied to the particles of $(NH_4)_2SO_4$ ($\rho_p = 1.77$ g/cm$^3$) and $NaNO_3$
($\rho_p = 2.26$ g/cm$^3$) with $D_a$ values of 250.0 nm, 350.0 nm, 450.0 nm and 550.0 nm.
Besides, to present the measurement uncertainty of the AAC, the $D_a$ values of these
PSL particles were measured to be $212.8 \pm 0.2$, $324.7 \pm 0.4$, $529.9 \pm 0.4$, and $767.5 \pm$
0.4 by the system of AAC- condensation particle counter (CPC), respectively. It shows
that the AAC has the deviations of 1.1%, 1.3%, 0.8%, and 0.7% for determining the $D_a$
values of the particles.
**2.3 Ambient sampling**
For field observations, the AAC-SPAMS system was deployed in Science and
Technology Enterprise Accelerator A2 Block, Guangzhou, China, to characterize the
$D_{ve}, \rho_e$ and chemical compositions of aerosol particles. The sampling inlet was hung 2.5
meters from the third floor (~12 m above ground level). Ambient aerosol particles were
introduced into the AAC through a 5 m long conductive silicone tube with an inner
diameter of 6 mm and a PM$_{2.5}$ cyclone inlet. The sampling flow from the PM$_{2.5}$ cyclone
inlet was 3 lpm, and the residence time in the conductive silicone tube was
approximately 5 seconds. Particles with the $D_a$ of 250.0, 350.0, 450.0, and 550.0 nm
were classified by the AAC. The sampling time for the particles of each $D_a$ was
approximately 10 minutes. From July 6[th] to 8[th], 2019, approximately 129,869 ionized
particles were obtained from nine rounds of measurement. The sampling details are
shown in Table S1. The number of ionized particles with the $D_a$ of 250.0, 350.0, 450.0,
and 550.0 nm is 35,609, 38,374, 31,910, and 23,976, respectively. The sampled
~100,000 particles are first classified by using an adaptive resonance theory neural
network (ART-2a) (Song et al., 1999) with a vigilance factor of 0.75, a learning rate of
0.05 and 20 iterations.

**2.4 Theoretical derivation of $D_{ve}$ and $\rho_e$ from $D_a$ and $D_{va}$**
In this study, the calculations of $D_{ve}$ and $\rho_e$ for unknown particles are theoretically
derived from $D_a$ and $D_{va}$. Combining Eqs. (2) and (3), we obtain the following Eq. (10):
$$C_c(D_a)\frac{D_a^2}{D_{va}} = D_{ve}C_c(D_{ve})\frac{\chi_v}{\chi_t} \tag{10}$$

Based on the approximation between $\chi_v$ and $\chi_t$ ($\chi_v \approx \chi_t = \chi_a$) (DeCarlo et al., 2004), Eq.
(10) becomes Eq. (11):
$$C_c(D_a)\frac{D_a^2}{D_{va}} = D_{ve}C_c(D_{ve}) \tag{11}$$

The Cunningham Slip Correction Factor is calculated by Eq. (12) (Peng and Bi, 2020):
$$C_c(D) = 1 + \frac{\lambda}{D}\left(A + B\cdot\exp\left(\frac{C\cdot D}{\lambda}\right)\right), \tag{12}$$

where $\lambda$ is the mean free path of the gas molecules, and $A$, $B$ and $C$ are empirically
determined constants specific to the analysis system. The values of $A$, $B$ and $C$ are 2.33,
0.966, and -0.498 provided by the manual of the AAC. Substituting Eq. (12) into Eq.
(11) obtains the Eq. (13).
$$\frac{D_a^2}{D_{va}} + \frac{D_a\cdot\lambda}{D_{va}}\left(A + B\cdot\exp\left(\frac{C\cdot D_a}{\lambda}\right)\right) = D_{ve} + \lambda\left(A + B\cdot\exp\left(\frac{C\cdot D_{ve}}{\lambda}\right)\right) \tag{13}$$

If the $D_a$ and $D_{va}$ of an unknown particle can be measured, its $D_{ve}$ could be calculated
according to Eq. (13). Finally, the $\rho_e$ value of the particles is calculated by the $D_{va}$ and
$D_{ve}$ values according to Eq. (14), which is obtained by combining Eq.(3) and Eq.(5):
$$\rho_e = \frac{\rho_p}{\chi_a} = \frac{D_{va}}{\rho_0\cdot D_{ve}} \tag{14}$$

Thus, we can obtain both the $D_{ve}$ and $\rho_e$ values of unknown particles based on the $D_a$
and $D_{va}$ values. Because the AAC and SPAMS instruments have the ability to determine
$D_a$ and $D_{va}$, the AAC-SPAMS system developed in this study can be used to obtain the
$D_{ve}$ and $\rho_e$ values for unknown particles.

## 3. Results and discussion

### 3.1 Verification of the AAC-SPAMS system to obtain $D_{ve}$ and $\rho_e$

The $D_{va}$ distribution of PSL particles with predefined $D_{ve}$ values after screened by the AAC is shown in Figure S1. We used Gaussian fitting to obtain the peak $D_{va}$ for each size PSL with an R-squared fitting coefficient ($R^2$) over 0.98. Each fitting has a full width at half maximum (FWHM) of 6.6%, 4.4%, 2.3% and 2.2%, and the corresponding peaks are 215.8 nm, 319.0 nm, 532.1 nm and 803.5 nm, respectively. Substituting the $D_a$ and $D_{va}$ values of PSL into Eq. (11), the measured $D_{ve}$ ($D_{ve,me}$) of PSL from AAC-SPAMS system is 203.6 nm, 309.7 nm, 511.6 nm and 737.2 nm, respectively (Figure 2a). Thus, the deviations between the theoretical $D_{ve}$ ($D_{ve,th}$) and $D_{ve,me}$ values are 0.3%, -0.1%, 0.3% and -0.4%, respectively. On the other hand, the measured $\rho_e$ ($\rho_{e,me}$) values of the particles, calculated from the $D_{va}$ and $D_{ve,me}$ values with Eq. (14), are 1.1 g/cm$^3$, 1.0 g/cm$^3$, 1.0 g/cm$^3$, and 1.1 g/cm$^3$, respectively (Figure 2b). Comparing to the theoretical $\rho_e$ ($\rho_{e,th}$) (i.e. 1.055 g/cm$^3$ of PSL particles), the deviations of $\rho_{e,me}$ are determined to be 4.3%, -5.2%, -5.2%, and 4.3%, respectively. That is, the deviations of $D_{ve,me}$ and $\rho_{e,me}$ obtained by the AAC-SPAMS system are within 1% and 6%, respectively.

### 3.2 Application of the AAC-SPAMS system for obtaining $D_{ve}$ and $\rho_e$ of $(NH_4)_2SO_4$ and $NaNO_3$

Figure S2 shows the $D_{va}$ distributions of $(NH_4)_2SO_4$ and $NaNO_3$ particles with $D_a$ values of 250.0, 350.0, 450.0, and 550.0 nm screened by the AAC. The $D_{va}$ peaks are

obtained by Gaussian fitting, with $R^2$ values over 0.93 and FWHM values ranging from
7.6% to 10.6%. The $(NH_4)_2SO_4$ particles have $D_{va}$ values of 300.0, 418.0, 551.1, and
695.1 nm (Figure S2), which correspond to particles possessing $D_{ve,me}$ values of 177.3,
254.4, 331.8, and 409.3 nm, respectively, according to Eq. (11). Substituting the values
of $D_{va}$ and $D_{ve,me}$ into Eq. (12), the $\rho_{e,me}$ values are 1.7, 1.6, 1.6, and 1.7 g/cm$^3$ (Figure
3a), respectively. Similarly, the selected $NaNO_3$ particles are determined to have $D_{va}$
values of 321.0, 454.9, 599.8, and 755.3 nm (Figure S2), corresponding to $D_{ve,me}$ values
of 150.1, 218.2, 287.0, and 355.9 nm, respectively. The $\rho_{e,me}$ values of the $NaNO_3$
particles are 2.2, 2.0, 2.0, and 2.1 g/cm$^3$ (Figure 3b), respectively. Figure 3 also shows
that the $\rho_{e,me}$ values of the $NaNO_3$ and $(NH_4)_2SO_4$ particles with different $D_a$ deviate
from their average values with the maximum of 5.9 % and 4.8%, respectively, which
are identical with the deviation for the $\rho_{e,me}$ of PSL particles. These deviations may be
derived from the calibration of particle $D_{va}$ from the SPAMS. While the R-square of
size calibration curve is 0.999, the curve of exponential function is found to slightly
deviate from the data points measured by SPAMS. For example, size calibration
function produces the deviation of -4.4% and 3.1% from the data points of 310 and 740
nm, respectively.
Taking the systematic error into account, the slight difference of the $\rho_{e,me}$ values for
the four sizes suggests that the $\rho_e$ of $(NH_4)_2SO_4$ and $NaNO_3$ particles is independent of
particle size from 250.0 nm to 550.0 nm. It is determined by the definition of effective
density used in this study, which keeps constant as long as the $\chi_a$ of the particles does
not change with particle size for pure compound. The average $\rho_{e,me}$ values of $(NH_4)_2SO_4$
and NaNO$_3$ particles are calculated to be $1.7 \pm 0.1$ and $2.1 \pm 0.1$ g/cm$^3$, which are lower
than the $\rho_p$ of (NH$_4$)$_2$SO$_4$ (1.77 g/cm$^3$) and NaNO$_3$ (2.27 g/cm$^3$). This is partly caused
by the $\chi_a$, which can be used to parameterize the morphology. According to Eq. (14),
the $\chi_a$ with different $D_a$ are calculated to be 1.04, 1.11, 1.11, and 1.04 for (NH$_4$)$_2$SO$_4$
particles and to be 1.03, 1.14, 1.14, and 1.08 for NaNO$_3$ particles. Thus, the average $\chi_a$
values of the (NH$_4$)$_2$SO$_4$ and NaNO$_3$ particles are determined to be $1.07 \pm 0.04$ and 1.10
$\pm 0.05$, respectively, indicating that these particles are aspherical.
The asphericity of (NH$_4$)$_2$SO$_4$ determined by AAC-SPAMS system is consistent with
the previous studies reporting that the $\chi_a$ of (NH$_4$)$_2$SO$_4$ were larger than the value of
1.03 (Zelenyuk et al., 2006; Beranek et al., 2012; Zhang et al., 2016a). However,
previous studies found that the NaNO$_3$ particles had different morphology. Zhang et al.
(2016a) observed that NaNO$_3$ had the $\chi_a$ of 1.09-1.13, while Hoffman et al. (2004) found
that NaNO$_3$ particle had a round droplet-like shape even at 15% RH, supported by the
consistence between the measured value of "anhydrous" droplet density and the
calculated value of "anhydrous" solution droplet (Zelenyuk et al., 2005). Eclectically,
Tang and Munkelwitz (1994) studied that most of the NaNO$_3$ particles crystallized
between 20% and 30% RH but some persisted down to 10% RH to keep solution
droplets. Notably, the spherical NaNO$_3$ particles at low RH observed by Hoffman et al.
(2004) were dried in the sticky carbon tape which might affect the phase transition of
droplet-like NaNO$_3$ particles. In this study, most NaNO$_3$ particles might crystallize
because the RH of the aerosol flow carrying the NaNO$_3$ particles was reduced to below
20% through the diffusion drying tube. The asphericity of the crystallized NaNO$_3$
particles is supported by their FWHM values of the $D_{va}$ distributions, which are
consistent with that of aspherical $(NH_4)_2SO_4$ (Figures S1 and S2).

**3.3 Application of the AAC-SPAMS system for measuring the chemically-resolved**
$D_{ve}$ **and** $\rho_e$

SPAMS can obtain information on the chemical composition of individual particles,

implying that the AAC-SPAMS system has the ability to simultaneously characterize
$D_{ve}$, $\rho_e$ and the chemical compositions of particles in real time. It is worth noting that
the freshly emitted soot particles exhibit the largest $\chi$ (~2.5) in the actual atmosphere
(Peng et al., 2016). It meets the upper limit for the approximation between the $\chi_t$ and $\chi_v$
(DeCarlo et al., 2004).

As an example, the AAC-SPAMS system was deployed in the field to obtain the

chemically-resolved $D_{ve}$ and $\rho_e$ values for unknown aerosol particles. The sampled
~100,000 particles are classified into eight major particle types with distinct chemical
composition: K-rich, EC-S, K-Na, Amine, EC-N-S, OC-N-S and OC-EC-N-S and
Metal-rich, representing 97% of the detected particle population. Details of the
chemical composition and number fraction of the eight types of particles are presented
in the Figure S3 and Figure S4, respectively, which are discussed in the Supporting
Information.

We used Gaussian fitting to obtain the $D_{va}$ peaks for each particle type with $D_a$ values

of 250.0 nm, 350.0 nm, 450.0 nm, and 550.0 nm. Then, we calculated the $D_{ve}$ values of
the atmospheric particles with Eq. (11). Table 1 presents the average $D_{ve}$ values of the
eight particle types, for which the standard deviation is calculated based on nine
samples. The average $D_{ve}$ at $D_a$ values of 250.0 nm, 350.0 nm, 450.0 nm, and 550.0 nm
shows wide ranges: from 188.5 nm to 200.8 nm, 271.9 nm to 295.7 nm, 342.5 nm to
428.9 nm, and 397.3 nm to 570.9 nm, respectively, which are caused by the different
chemical composition. The result indicates that particles with significantly different $D_{ve}$
might possess the same $D_a$. Furthermore, the large standard deviation of $D_{ve}$, such as
21.9 nm for K-Na at 250.0 nm, 32.3 nm for OC-EC-N-S at 350.0 nm, and 44.3 nm for
OC-N-S at 450.0 nm, indicates that the $D_{ve}$ of particles is remarkably different even for
particles with the same type and same $D_a$.
According to $D_{ve}$ and $D_{va}$, we calculated the $\rho_e$ of each particle type by Eq. (12).
Figure 4 shows the variations of the $\rho_e$ with $D_{ve}$ for nine particle samples. For pure
compounds, such as $(NH_4)_2SO_4$ and $NaNO_3$ particle, $\rho_e$ theoretically does not change
with particle size. However, the sampled particles have experienced complex
atmospheric processes. Therefore, $\rho_e$ has a very wide distribution for each type of
particles with a similar $D_{ve}$. Specifically, the $\rho_e$ of K-Na increases with $D_{ve}$, while the $\rho_e$
of OC-N-S and OC-EC-N-S decreases with $D_{ve}$, which may be influenced by the
particle shape or the material density. Additionally, the average $\rho_e$ of each type of
particle is in the order from small to large: $1.2 \pm 0.2$ g/cm$^3$ for OC-EC-N-S, $1.3 \pm 0.2$
g/cm$^3$ for OC-N-S, $1.4 \pm 0.1$ g/cm$^3$ for K-rich, $1.4 \pm 0.1$ g/cm$^3$ for Amine, $1.5 \pm 0.1$
g/cm$^3$ for EC-N-S, $1.5 \pm 0.1$ g/cm$^3$ for EC-S, $1.6 \pm 0.1$ g/cm$^3$ for K-Na and $1.6 \pm 0.1$
g/cm$^3$ for Metal-rich. It is reasonable to find that the average $\rho_e$ of internally mixed
particles distributes in the range of their material densities ($\rho_m$). For instance, the OC-
EC-N-S, OC-N-S, K-rich, and Amine particles, mainly comprised of internally mixed
sulfate and organics, have the average $\rho_e$ between that of sulfate with $\rho_m$ of 1.77 g/cm$^3$
and organic aerosols with $\rho_m$ of 1.2 g/cm$^3$ (Cross et al., 2007).

**4. Conclusion**

We develop an AAC-SPAMS system to first achieve the measurement of the $D_{ve}$ and
$\rho_e$ (defined as the ratio of $\rho_p$ to $\chi$) of the aspherical particles through characterizing their
$D_a$ and $D_{va}$. The reliability of the AAC-SPAMS system is verified by accurately
measuring the $D_{ve}$ and $\rho_e$ of PSL. Applying the AAC-SPAMS system to determine the
$D_{ve}$ and $\rho_e$ of $(NH_4)_2SO_4$ and $NaNO_3$ particles shows that these particles are aspherical
and their $\rho_e$ are independent of particle size. Coupled with the ability of SPAMS to
characterize the chemical composition of individual particles, the AAC-SPAMS system
was demonstrated to be capable of characterizing the $D_{ve}$, $\rho_e$ ($\rho_p/\chi$) and chemical
compositions of atmospheric particles simultaneously, showing the potential
application of this system in field observations. The approach achieves the
measurement of chemically-resolved $D_{ve}$ and $\rho_e$ ($\rho_p/\chi$), and provides the possibility to
determine their quantitative relationship with other particle properties, which would be
benefit for further reduction of the uncertainty associated with the effects of particles
on air quality, human health and radiative forcing.

***Data availability.*** Data in this study is available at https://github.com/longer1217/All-
figures-data.

*Author contributions.* The idea for the study was conceived by LP and GHZ. All experiments were performed by LP with the assistance of LL. LP wrote the paper which was reviewed by GHZ and XHB. All co-authors discussed the results and commented on the manuscript.

**Competing interests.** The authors declare they have no conflict of interest.

**Acknowledgment**

This work was supported by the National Nature Science Foundation of China (41775124 and 41877307), Natural Science Foundation of Guangdong Province (2019B151502022), and the Guangdong Foundation for the Program of Science and Technology Research (2019B121205006 and 2017B030314057). The authors also gratefully acknowledge Cambustion Ltd., UK for providing the AAC and Hexin Analytical Instrument Co., Ltd., China for providing the SPAMS.

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

**Table 1.** The measured mean $D_{ve}$ and its standard deviation for the eight particle types at $D_a$ values
of 250.0 nm, 350.0 nm, 450.0 nm, and 550.0 nm from nine round measurement.

| $D_a$ (nm) | K-rich | EC-S | K-Na | Amine |
|---|---|---|---|---|
| **250.0** | $193.1 \pm 8.2$ | $192.2 \pm 8.1$ | $193.8 \pm 21.9$ | $190.6 \pm 4.6$ |
| **350.0** | $284.0 \pm 28.4$ | $280.8 \pm 9.3$ | $271.9 \pm 18.0$ | $284.8 \pm 18.2$ |
| **450.0** | $364.7 \pm 21.1$ | $357.8 \pm 6.9$ | $342.5 \pm 7.3$ | $367.9 \pm 9.7$ |
| **550.0** | $416.6 \pm 28.3$ | $439.5 \pm 5.4$ | $397.3 \pm 29.7$ | $442.5 \pm 7.4$ |
| $D_a$ (nm) | EC-N-S | OC-N-S | OC-EC-N-S | Metal-rich |
| **250.0** | $188.5 \pm 5.9$ | $200.8 \pm 17.9$ | $195.4 \pm 8.9$ | $189.0 \pm 6.7$ |
| **350.0** | $281.3 \pm 9.3$ | $295.7 \pm 29.8$ | $294.0 \pm 32.3$ | $277.0 \pm 9.1$ |
| **450.0** | $358.0 \pm 5.8$ | $398.3 \pm 44.3$ | $428.9 \pm 24.0$ | $342.9 \pm 10.0$ |
| **550.0** | $453.2 \pm 16.4$ | $547.4 \pm 14.7$ | $570.9$ | $407.4 \pm 14.5$ |


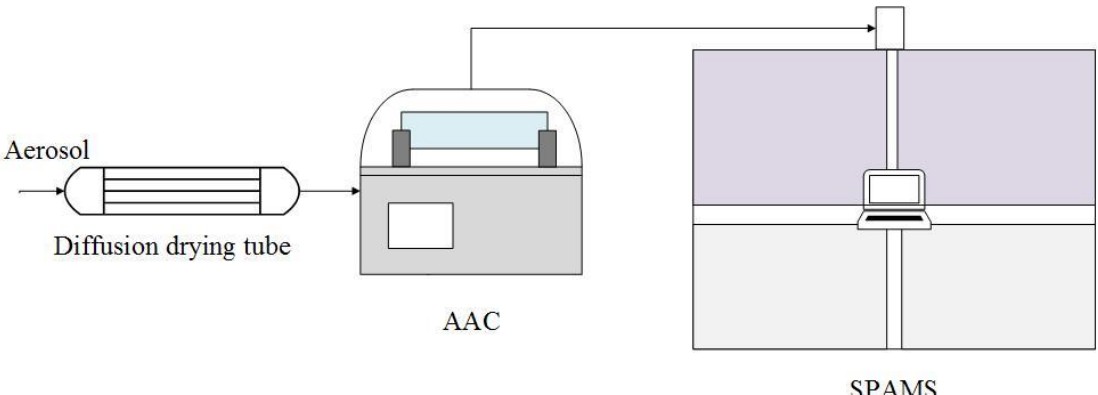


**Figure 1.** Schematic diagram of the AAC-SPAMS system (0.3 lpm). The diffusion drying tube is

filled with orange silica gel, which reduces the RH to 5-15%.

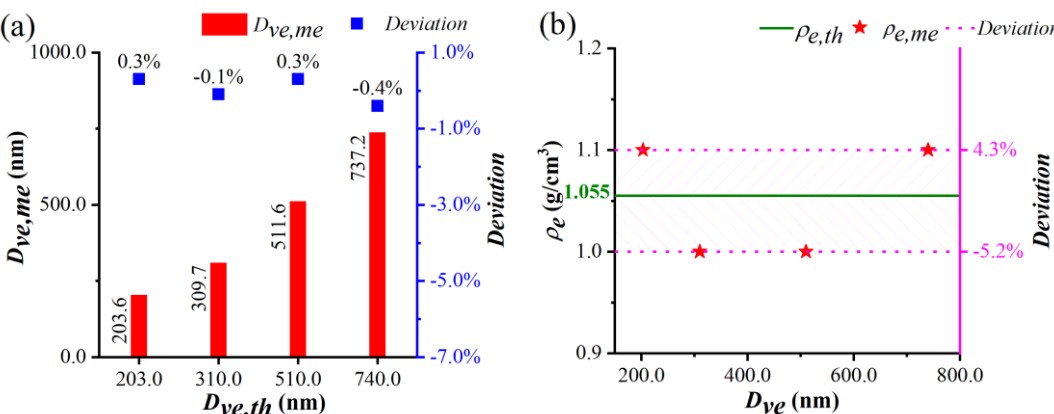



**Figure 2.** (a) Comparison between the measured $D_{ve}$ ($D_{ve,me}$) and the theoretical $D_{ve}$ ($D_{ve,th}$) of the
PSL particles. (b) Comparison between the measured $\rho_e$ ($\rho_{e,me}$) and the theoretical $\rho_e$ ($\rho_{e,th}$) of the
PSL particles.

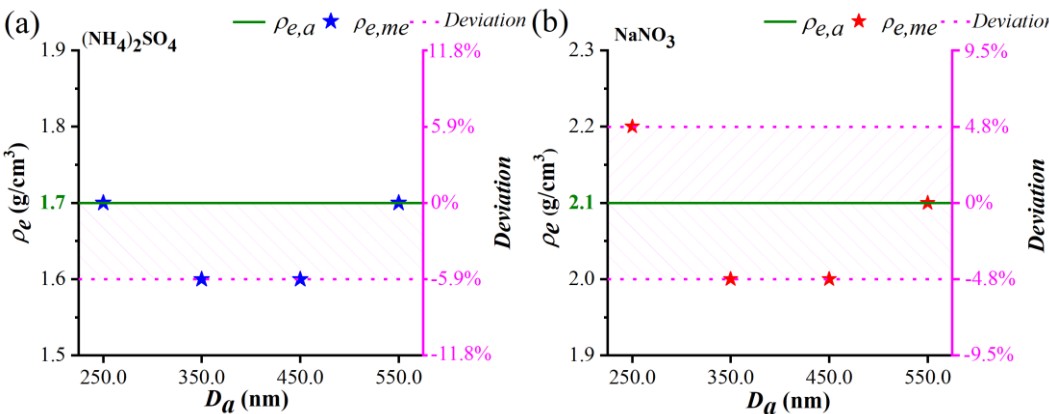



**Figure 3.** (a) Comparison between the measured $\rho_e$ ($\rho_{e,me}$) and average $\rho_e$ ($\rho_{e,a}$) values of the
$(NH_4)_2SO_4$ particles. (b) Comparison between the measured $\rho_e$ ($\rho_{e,me}$) and average $\rho_e$ ($\rho_{e,a}$) values of
the $NaNO_3$ particles.

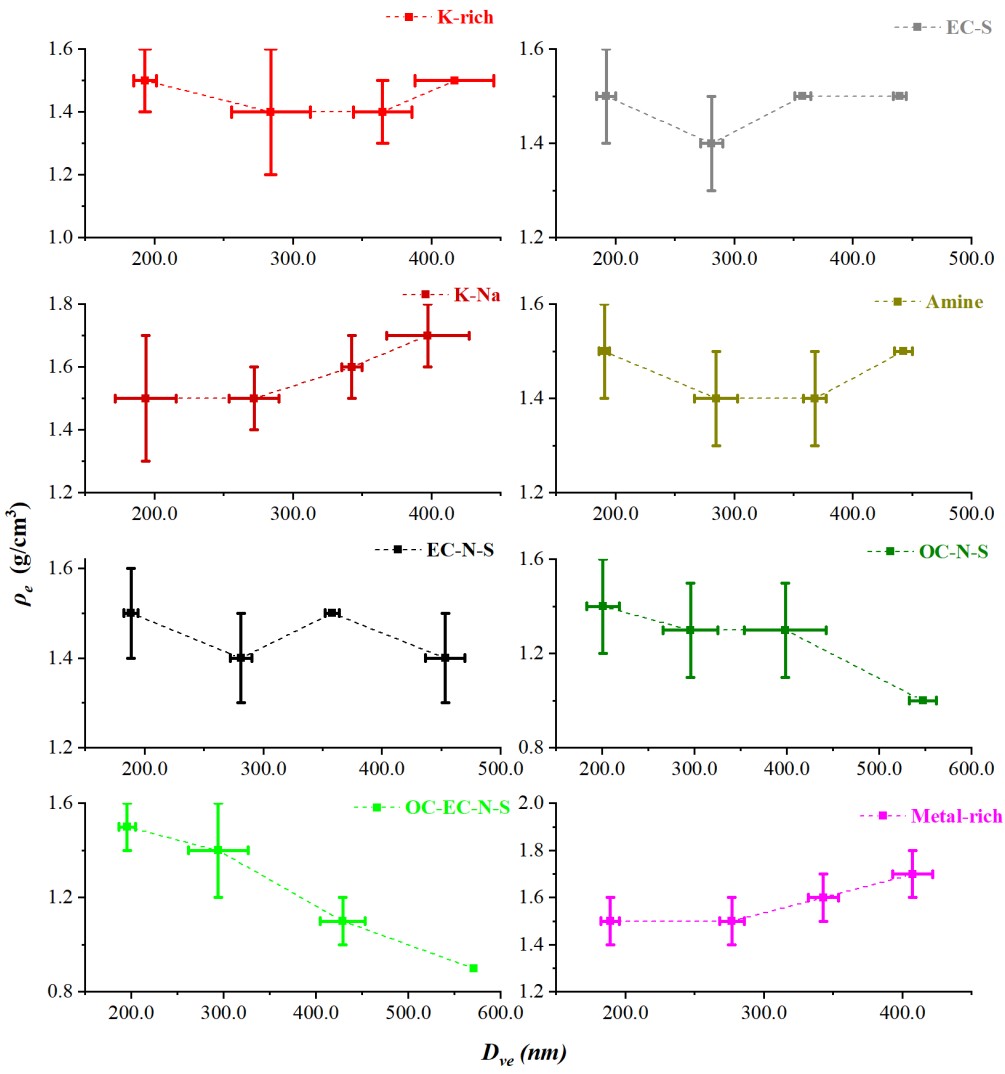

**Figure 4.** Variation in $\rho_e$ of the eight particle types with $D_{ve}$. The solid lines represent the rang of
the $\rho_e$ and $D_{ve}$ measured from nine rounds, and the data points stand for the average values.