# Peer review of "Long Peng1,2, Lei Li4, Guohua Zhang1, 3\*, Xubing Du4, Xinming Wang1, 3, Ping'an"

_Atmospheric Chemistry and Physics, 2020_

## Referee Comment (RC1) · Johannes Schneider (Referee) · 30 Nov 2020

Review to "Measurement of chemically resolved volume equivalent diameter and effective density of particles by AAC - SPAMS" by Peng et al, submitted as technical note to ACP

This manuscript presents an interesting application of an aerodynamic aerosol classifier (AAC) and a single particle mass spectrometer (SPMS) to obtain particle properties like effective density and shape.

The idea of using aerosol mass spectrometers that measure the vacuum aerodynamic

diameter in combination with other size measurements to obtain shape and density of aerosol particles is not new. In 2006, two papers where published on this topic: one by Zelenyuk et al. (ZEL2006), the other by my group, Schneider et al., 2006 (SCH2006). While the authors of the present manuscript reference ZEL2006, they may have overlooked our paper, because we used an Aerodyne AMS and not a laser ablation SPMS.

Both ZEL2006 and SCH2006 used the mobility diameter measured by a differential mobility analyzer (DMA) and not the AAC that measures the aerodynamic diameter, but the basic idea is the same. In SCH2006, we already presented the equation for rho_eff_II (your Equ. 8). Thus, please give proper credit to our work.

In general, this technical note is appropriate for ACP. The combination of single particle results (via the clustering algorithms), effective density and shape factors is interesting and is worth to be exploited further.

However, I have some further important comments that have to be addressed before this paper can be published:

Lines 68-76: You rely here on DeCarlo et al., 2004, but the numbering of the effective density is different. rho_e_I is the same, but you changed rho_e_II and rho_e_III compared to deCarlo et al. Please use the same numbering to avoid confusion. Please also refer to Hand et al., 2002 who introduced your rho_e_III (which in de Carlo et al. is termed rho_eff_II).

Line 80, equation 8: rho_0 is wrong here, needs to be deleted to get the units right.

Line 81: "The detailed derivation will be presented in a separate paper". The derivation of Equ. 8 was given in deCarlo et al., 2004, and also in SCH2006, so please give proper reference here, and for completeness, give the derivation of Equ. 7 here as well.

Line 157, Equ 11: To calculate D_ve, you need the Cunningham slip correction values here. How are they obtained? The differences between D_ve and D_a (e.g. Fig. S2)

are rather large, so the Cunningham correction can not be neglected.

Line 161: I think that Equ. 12 results from combining Equ. 3 and Equ. 6. Correct?

Line 176: Please state clearly how D_ve,th was calculated. I would assume that again the Cunningham correction is needed to do this?

Line 180 – 182: Please clarify that for spherical particles like PSL, rho_e = rho (see deCarlo et al., Equ [43] and thereafter).

Line 206-209: If different definitions of the effective densities are used, the statement "This pattern is divergent with the previous studies, which showed that effective density decreased as the size increasing" has to be removed or at least reworded.

Line 259-260: Please refer to the respective Figures in the Supporting Information.

Line 275-280: rho_e = rho_p / shape factor (Equ. 6). Thus, either the density of the particle material is a function of size, or the particle shape factor (or both). I think you can not rule out that the material density changes with size. SPMS is not quantitative, so particles of the same cluster type may have different quantitative composition (e.g. the ratio OC/EC or organic/inorganic). Thus, you can't tell whether the changing rho_e with size is an effect of shape or composition.

Supplement: Fig. S2b) is missing.

References:

Schneider, J., S. Weimer, F. Drewnick, S. Borrmann, G. Helas, P. Gwaze, O. Schmid, M. O. Andreae, and U. Kirchner: Mass spectrometric analysis and aerodynamic properties of various types of combustion-related aerosol particles, Int. J. Mass. Spec., 258, 37–49, https://doi.org/10.1016/j.ijms.2006.07.008, 2006.

Zelenyuk, A., Cai, Y., and Imre, D.: From Agglomerates of Spheres to Irregularly Shaped Particles: Determination of Dynamic Shape Factors from Measurements of Mobility and Vacuum Aerodynamic Diameters, Aerosol Sci. Tech., 40, 197–217,

https://doi.org/10.1080/02786820500529406, 2006.

DeCarlo, P. F., Slowik, J. G., Worsnop, D. R., Davidovits, P., and Jimenez, J. L.: Particle morphology and density characterization by combined mobility and aerodynamic diameter measurements. Part 1: Theory, Aerosol Sci. Technol., 38, 1185-1205, 2004.

Hand, J. L., Kreidenweis, S. M., Kreisberg, N., Hering, S., Stolzenburg, M., Dick, W., and McMurry, P.H.: Comparisons of Aerosol Properties Measured by Impactors and Light Scattering from Individual Particles: Refractive Index, Number and Volume Concentrations, and Size Distributions, Atmos. Environ. 36(11):1853–1861. https://doi.org/10.1016/S1352-2310(02)00103-6, 2006

---

## Referee Comment (RC2) · Johannes Schneider (Referee) · 30 Nov 2020

I am sorry, but I gave the wrong reference Hand et al. 2002.

The correct one is this:

Jenny L. Hand & Sonia M. Kreidenweis (2002) A New Method for RetrievingParticle Refractive Index and Effective Density from Aerosol Size Distribution Data, AerosolScience & Technology, 36:10, 1012-1026, DOI: 10.1080/02786820290092276

Johannes Schneider

---

## Referee Comment (RC3) · Anonymous Referee #2 · 5 Dec 2020

Peng et al present results from coupling an aerodynamic aerosol classifier (AAC) with a single particle aerosol mass spectrometer (SPAMS) to determine individual particle effective density and shape. While this is a worthwhile endeavor, the manuscript misrepresents the prior work on the topic (deriving effective density using a size measurement prior to a single-particle mass spectrometer, which was shown several times in the 2000s). I agree with Johannes Schneider's review and provide additional comments here. A main additional point is that the results need to include propagated measurement error reporting, in the figures and text, for accurate comparisons to theoretical/manufacturer values and to support size-dependent trends, or the lack thereof.

As the prior review stated, deriving effective density using a size measurement prior to a single-particle mass spectrometer is not new, and this prior work needs to be discussed in a dedicated added section in the introduction to properly place the current work in context. There are statements, such as those on lines 90, 150-151, and 298-299, that give the impression that this is the first work to measure chemically-resolved particle effective density, and therefore, this phrasing needs to be revised. The lack of discussion of this prior work is surprisingly given the author's prior paper, which is cited (Zhang et al. 2016, Sci. China Earth Sci., Measurement of aerosol effective density by single particle mass spectrometry). The introduction of the current manuscript only mentions one prior paper (Moffet and Prather 2005) that derived effective density using an aerosol mass spectrometer (and that work was using scattering signals) and does not discuss prior derivations of shape factors. In addition to Zelenyuk et al. (2006) and Schneider et al. (2006) referenced in the previous review, additional work, not cited in the current paper, includes: - Murphy et al. (2004), J. Aerosol Sci., Particle density inferred from simultaneous optical and aerodynamic diameters sorted by composition - Spencer et al. (2007), Environ. Sci. Technol., Simultaneous Measurement of the Effective Density and Chemical Composition of Ambient Aerosol Particles - Slowik et al. (2007), Aerosol Sci. Technol., Measurements of morphology changes in fractal soot particles using coating and denuding experiments: Implications for optical absorption and atmospheric lifetime - Zelenyuk et al. (2008), Analytical Chem., Simultaneous measurements of individual ambient particle size, composition, effective density, and hygroscopicity - Zelenyuk et al. (2008), Environ. Sci. Technol., A new real-time method for determining particles' sphericity and density: Application to secondary organic aerosol formed by ozonolysis of a-pinene - Zelenyuk et al. (2008), J. Phys. Chem. A, "Depth-profiling" and quantitative characterization of the size, composition, shape, density, and morphology of fine particles with SPLAT, a single-particle mass spectrometer - Alexander et al. (2016), Aerosol Sci. Technol., Measurement of size-dependent dynamic shape factors of quartz particles in two flow regimes This may not be a complete list, and therefore, the authors need to do a thorough literature search. In addition to addition to the Introduction, this prior work should also be discussed/compared to in the Results & Discussion. For example, paragraph 2 on page 11 does a good job of comparing to and discussing previous literature, and the manuscript would benefit from this type of comparison and discussion at other points in the manuscript.

The manufacturer reported uncertainties on the PSL sizes need to be reported in the Methods, given the goal of measurement comparison to these values, and then they need to be included when calculating "discrepancies" with the measurements. It is quite possible that the measurement comparison is well within the expected uncertainties. Currently the sizes are stated at 0.X nm accuracy, but my understanding is that the manufactured PSLs are not this monodisperse. For example, a quick look at the manufacturer website suggests that the 203 nm PSLs are provided at +/- 5 nm. Given the direct comparison in the Results to these sizes, the reported accuracy by the manufacturer is key and needs to be reported. In addition, for the ammonium sulfate, ammonium nitrate, and ambient particles, what is the error in the size selection by the AAC? This is also critical to the method uncertainty. The AAC brochure says that size resolution depends on the sheath to sample flow ratio, so this should also be reported in the methods. Throughout the Results and Discussion text and associated figures, the full measurement uncertainty (that takes into account the width of the size distribution and that it is not monodisperse to the tenth of a nanometer, as implied by reporting values to 0.X nm) needs to be calculated and included in the difference ("discrepancy") calculations. These error bars are particularly needed in Figures 2 and 3 (similar to the inclusion in Figure 4) and in the assessment of any size dependence of effective density. The Figure 4 caption should also state the origin of the error bars shown. Further, on lines 202-203, it is stated that ". . .in the SPAMS [the] size calibration curve possesses the systematic error." However, this systematic error is not stated or shown (nor is it's origin explained). Further, on Line 206, it is stated that the size-dependent pattern observed is "divergent with the previous studies", but without inclusion of measurement uncertainty, any "pattern" or trend cannot be assessed. Further, often too

many decimal places are reported in the manuscript, beyond the appropriate number of significant figures, and this should be evaluated once error is calculated.

In the methods section, Section 2.2 should be separated into laboratory experiments and ambient sampling, for improved clarity. Information should be provided about the diffusion drying tube shown in Figure 1, especially since the water content of the particles is key to the results. The description of the clustering methodology should be moved from the Results to the Methods and expanded. The location and dates of ambient sampling also need to be provided, as well as the actual number of ambient particles measured at each size selected.

Additional comments:

Lines 209-211: This sentence is not clear and makes mention of a separate paper being written on the topic of effective density and size-dependent evaluation, but it is not clear why that isn't included here or how it is different.

Line 239: By "different", do you mean "wider"?

Lines 255-256: Add a reference to this methods sentence and move to the Methods section.

Line 281 and Table 1: Note that error should only be reported with 1 significant figure.

Table 1 caption: State what the error corresponds to here.

Figure 1 caption: Add flow rates to the figure to make it more informative.

Figure S2b is missing.
* * *

---

## Author Comment (AC2) · 27 Jan 2021

**Response to comments**

Response: We thank the reviewers for thoughtful suggestions and constructive criticism that have helped us improve our manuscript. Below we have detailed responses and resulting edits to all of the reviews' comments. The review comments are listed in italics and black, followed by our responses in normal font and blue. To make it clear, the contents in revised manuscript are presented in quotes and normal font. Reference to line numbers are added to the revised manuscript.

**Referee#2:**

*General Comments: Peng et al present results from coupling an aerodynamic aerosol classifier (AAC) with a single particle aerosol mass spectrometer (SPAMS) to determine individual particle effective density and shape. While this is a worthwhile endeavor, the manuscript misrepresents the prior work on the topic (deriving effective density using a size measurement prior to a single-particle mass spectrometer, which was shown several times in the 2000s). I agree with Johannes Schneider's review and provide additional comments here. A main additional point is that the results need to include propagated measurement error reporting, in the figures and text, for accurate comparisons to theoretical/manufacturer values and to support size-dependent trends, or the lack thereof.*

We thank the reviewer for his/her substantial efforts for scrutinizing the manuscript and giving valuable comments and constructive suggestions for improving our manuscript.

1) *As the prior review stated, deriving effective density using a size measurement prior to a single-particle mass spectrometer is not new, and this prior work needs to be discussed in a dedicated added section in the introduction to properly place the current work in context. There are statements, such as those on lines 90, 150-151, and 298-299, that give the impression that this is the first work to measure chemically-resolved particle effective density, and therefore, this phrasing needs to be revised. The lack of*

*discussion of this prior work is surprisingly given the author's prior paper, which is cited (Zhang et al. 2016, Sci. China Earth Sci., Measurement of aerosol effective density by single particle mass spectrometry). The introduction of the current manuscript only mentions one prior paper (Moffet and Prather 2005) that derived effective density using an aerosol mass spectrometer (and that work was using scattering signals) and does not discuss prior derivations of shape factors. In addition to Zelenyuk et al. (2006) and Schneider et al. (2006) referenced in the previous review, additional work, not cited in the current paper, includes:*

*- Murphy et al. (2004), J. Aerosol Sci., Particle density inferred from simultaneous optical and aerodynamic diameters sorted by composition*

*- Spencer et al. (2007), Environ. Sci. Technol., Simultaneous Measurement of the Effective Density and Chemical Composition of Ambient Aerosol Particles*

*- Slowik et al. (2007), Aerosol Sci. Technol., Measurements of morphology changes in fractal soot particles using coating and denuding experiments: Implications for optical absorption and atmospheric lifetime*

*- Zelenyuk et al. (2008), Analytical Chem., Simultaneous measurements of individual ambient particle size, composition, effective density, and hygroscopicity*

*- Zelenyuk et al. (2008), Environ. Sci. Technol., A new real-time method for determining particles' sphericity and density: Application to secondary organic aerosol formed by ozonolysis of a-pinene*

*- Zelenyuk et al. (2008), J. Phys. Chem. A, "Depth-profiling" and quantitative characterization of the size, composition, shape, density, and morphology of fine particles with SPLAT, a single-particle mass spectrometer*

*- Alexander et al. (2016), Aerosol Sci. Technol., Measurement of size-dependent dynamic shape factors of quartz particles in two flow regimes.*

*This may not be a complete list, and therefore, the authors need to do a thorough literature search. In addition to addition to the Introduction, this prior work should also be discussed/compared to in the Results & Discussion. For example, paragraph 2 on page 11 does a good job of comparing to and discussing previous literature, and the manuscript would benefit from this type of comparison and discussion at other points*

*in the manuscript.*

Response: Thanks for your comments and presenting the references about the measurement techniques of the effective density. As suggested, the reference of Moffet and Prather (2005), Moffet et al. (2008), and Zhang et al. (2016a) have been supplemented and summarized in the Introduction. As noted by these studies, the achievements of the $\rho_e^{II}$ are only applicable to the spherical particles. Therefore, we may conclude that it is the first time to achieve the measurement of the $\rho_e^{II}$ and volume equivalent diameter for aspheric particles. Please refer to Lines 87-94:

"Previously, $\rho_e^{II}$ and the real part in the refractive index ($n$) can be retrieved from a fitting procedure that compares the measured light-scattering intensity of particles ($R_{meas}$) to the theoretical values ($R_{theory,test}$) calculated by a series of $n$ and $\rho_e^{II}$ values. Moffet and Prather (2005) successfully obtained $\rho_e^{II}$ for spherical particles by single particle mass spectrometry. However, subject to the accuracy of Mie theory for the aspherical particles, dry NaCl and calcium-rich dust particles were failed to fit the $R_{theory,test}$ well to $R_{meas}$ (Moffet et al., 2008). Similarly, Zhang et al. (2016a) failed to simultaneously retrieve $\rho_e^{II}$ and $n$ for $(NH_4)_2SO_4$ and $NaNO_3$ particles."

However, Other references were not included as we focused on the measurement of $\rho_e^{II}$ in the manuscript. We emphasized that three definitions of effective density should be considered as three different properties for the particles in essence. Their final expressions are presented as follows:

$$\rho_e^{I} = \frac{\rho}{\chi^3} \cdot \left(\frac{C_c(D_{ve})}{C_c(D_m)}\right)^3 \tag{1}$$

$$\rho_e^{II} = \frac{\rho_p}{\chi} \tag{2}$$

$$\rho_e^{III} = \rho \cdot \frac{C_c(D_{ve})}{\chi^2 \cdot C_c(D_m)} \tag{3}$$

where $C_c(D)$ is the Cunningham slip correction factor. The specific difference for three effective densities is shown in the following example: for soot particle with $\chi$ of 2.5, $\rho$ of 1.80 g/cm³ and $D_m$ of 550.0 nm, the values of $\rho_e^{I}$, $\rho_e^{II}$, $\rho_e^{III}$ are calculated to be 0.22, 0.72, and 0.36 g/cm³, respectively. Such a big gap suggests that it may not be appropriate to compare the three definitions of effective density together, as also suggested by the first reviewer.

*2) The manufacturer reported uncertainties on the PSL sizes need to be reported in the Methods, given the goal of measurement comparison to these values, and then they need to be included when calculating "discrepancies" with the measurements. It is quite possible that the measurement comparison is well within the expected uncertainties. Currently the sizes are stated at 0.X nm accuracy, but my understanding is that the manufactured PSLs are not this monodisperse. For example, a quick look at the manufacturer website suggests that the 203 nm PSLs are provided at +/- 5 nm. Given the direct comparison in the Results to these sizes, the reported accuracy by the manufacturer is key and needs to be reported.*

Response: We agree with the comments. We have added the uncertainties of the PSL sizes determined by Scanning Mobility Particles Sizer (Model 3938, TSI Inc., USA). Please refer to Lines 139-142: "Dried spherical PSL (Nanosphere Size Standards, Duke Scientific Corp., Palo Alto) ($\rho_p$ = 1.055 g/cm$^3$ and $\chi$ = 1.0) with $D_{ve}$ values of 203.0 ± 5.0 nm, 310.0 ± 6.0 nm, 510.0 ± 5.0 nm, and 740.0 ± 6.0 nm were used in the AAC-SPAMS system, and the $D_{ve}$ was verified by Scanning Mobility Particles Sizer (Model 3938, TSI Inc., USA)."

*In addition, for the ammonium sulfate, ammonium nitrate, and ambient particles, what is the error in the size selection by the AAC? This is also critical to the method uncertainty. The AAC brochure says that size resolution depends on the sheath to sample flow ratio, so this should also be reported in the methods. Throughout the Results and Discussion text and associated figures, the full measurement uncertainty (that takes into account the width of the size distribution and that it is not monodisperse to the tenth of a nanometer, as implied by reporting values to 0.X nm) needs to be calculated and included in the difference ("discrepancy") calculations. These error bars are particularly needed in Figures 2 and 3 (similar to the inclusion in Figure 4) and in the assessment of any size dependence of effective density. The Figure 4 caption should also state the origin of the error bars shown.*

Response: Thanks for the comments. The sample flow was 0.3 lpm in the setup of AAC-

SPAMS, and the value of Size Resolution Parameter (Rs) of the AAC was set as 40 (the Manual of AAC recommends its value ranges from 8 to 80 when the sample flow is 0.3 lpm), which has been added to the Experimental Section. Please refer to Lines 124-125: "The exhaust flow from the AAC was about 0.3 lpm, and the Size Resolution Parameter (Rs) of the AAC was set as 40." We know that the values for the size of PSL particles and the $D_{va}$ measured by SPAMS do not achieve the precision to 0.X nm. However, the size precision can be achieved to be lower than 1 nm when the AAC is applied.

We have revised the significant figures for effective density in the revised manuscript. Considering the precision of the PSL size is less than that of the instrument of AAC, the discrepancy between the measured value (from AAC-SPAMS) and the true value (the size and density of PSL) is used to represent the measurement uncertainty, which is presented in Figure 2 (deviations of size are 0.3%, -0.1%, 0.3% and -0.4%; deviations of density are 4.3%, -5.2%, -5.2%, and 4.3%). The meanings of the solid lines and the data points have been added in the Figure 4 caption, please refer to Lines 466-467: "**Figure 4.** Variation in $\rho_e$ of the eight particle types with $D_{ve}$. The solid lines represent the standard deviation of the $\rho_e$ and $D_{ve}$ measured from nine rounds, and the data points stand for the average values."

*Further, on lines 202-203, it is stated that ": : :in the SPAMS [the] size calibration curve possesses the systematic error." However, this systematic error is not stated or shown (nor is it's origin explained). Further, on Line 206, it is stated that the size-dependent pattern observed is "divergent with the previous studies", but without inclusion of measurement uncertainty, any "pattern" or trend cannot be assessed. Further, often too many decimal places are reported in the manuscript, beyond the appropriate number of significant figures, and this should be evaluated once error is calculated.*

Response: Thanks for the comments. The systematic error of size calibration curve has been stated accordingly. Please refer to Lines 224-232: "Figure 3 also shows that the $\rho_{e,me}$ values of the NaNO$_3$ and (NH$_4$)$_2$SO$_4$ particles at four size deviate from their

average values with the maximum of 5.9 % and 4.8%, respectively, which are identical with the deviation for the $\rho_{e,me}$ of PSL particles. These deviations may be derived from the calibration of particle $D_{va}$ from the SPAMS. While the R-square of size calibration curve is 0.999, the curve of exponential function is found to slightly deviate from the data points measured by SPAMS. For example, size calibration function has the deviation of -4.4% and 3.1% from the data points of 310 and 740 nm, respectively. "

3) *In the methods section, Section 2.2 should be separated into laboratory experiments and ambient sampling, for improved clarity. Information should be provided about the diffusion drying tube shown in Figure 1, especially since the water content of the particles is key to the results. The description of the clustering methodology should be moved from the Results to the Methods and expanded. The location and dates of ambient sampling also need to be provided, as well as the actual number of ambient particles measured at each size selected.*

Response: Thanks for the comments and the constructive suggestion. Section 2.2 has been separated into two section, and the location and dates of ambient sampling and the actual number of ambient particles measured at each size selected have been supplemented. Besides, the description of the clustering methodology has been moved from the Results to the Methods, and the corresponding references have been added. Please refer to Lines 138-162:

**"2.2 Laboratory experiments**

Dried spherical PSL (Nanosphere Size Standards, Duke Scientific Corp., Palo Alto) ($\rho_p = 1.055$ g/cm$^3$ and $\chi = 1.0$) with $D_{ve}$ values of $203.0 \pm 5.0$ nm, $310.0 \pm 6.0$ nm, $510.0 \pm 5.0$ nm, and $740.0 \pm 6.0$ nm were used in the AAC-SPAMS system, and the $D_{ve}$ was verified by Scanning Mobility Particles Sizer (Model 3938, TSI Inc., USA). The PSL particles were first classified by AAC, and then their $D_{va}$ values were obtained by the SPAMS. ACC-SPAMS was also applied to the particles of $(NH_4)_2SO_4$ ($\rho_p = 1.77$ g/cm$^3$) and $NaNO_3$ ($\rho_p = 2.26$ g/cm$^3$) with $D_a$ values of 250.0 nm, 350.0 nm, 450.0 nm and 550.0 nm.

**2.3 Ambient sampling**

For field observations, the AAC-SPAMS system was deployed in science and technology enterprise accelerator A2 Block, Guangzhou, China, to characterize the $D_{ve}$, $\rho_e$ and chemical compositions of aerosol particles. The sampling inlet was hung 2.5 meters from the third floor (~12 m above ground level). Ambient aerosol particles were introduced into the AAC through a 5 m long conductive silicone tube with an inner diameter of 6 mm and a PM$_{2.5}$ cyclone inlet. The overall sampling flow was 3 lpm, and the residence time was approximately 5 seconds. Sampled particles were classified by the AAC as one of four $D_a$: 250.0 nm, 350.0 nm, 450.0 nm and 550.0 nm. The sampling time for the particles of each $D_a$ was approximately 10 minutes. From July 6[th] to 8[th], 2019, approximately 129,869 ionized particles were obtained from nine rounds of measurement. The sampling details are shown in Table S1. The number of ionized particles with the $D_a$ of 250.0, 350.0, 450.0, and 550.0 nm is about 35,609, 38,374, 31,910, and 23,976, respectively. The sampled ~100,000 particles are first classified by using an adaptive resonance theory neural network (ART-2a) (Song et al., 1999) with a vigilance factor of 0.75, a learning rate of 0.05 and 20 iterations."

The information of the diffusion drying tube and the range of RH at outlet have been added in the Figure 1 caption:

Lines 452-453: "**Figure 1.** Schematic diagram of the AAC-SPAMS system (0.3 lpm). The diffusion drying tube is filled with orange silica gel, which reduces the RH to 5-15%."

4) *Lies 209-211: This sentence is not clear and makes mention of a separate paper being written on the topic of effective density and size-dependent evaluation, but it is not clear why that isn't included here or how it is different.*

Response: This sentence has been deleted in the manuscript. The difference among the three effective density has been illustrated in the Response for the first comment:

"The exact difference for three effective densities is shown in the following example: for soot particle with $\chi$ of 2.5, $\rho$ of 1.80 g/cm$^3$ and $D_m$ of 550.0 nm, the values of $\rho_e^I$,

$\rho_e^{II}$, $\rho_e^{III}$ are calculated to be 0.22, 0.72, and 0.36 g/cm$^3$, respectively." Besides, according to the result that soot particle with $\chi$ of 2.5, $\rho$ of 1.80 g/cm$^3$ and $D_m$ of 40.0 nm has the $\rho_e^{I}$, $\rho_e^{II}$, and $\rho_e^{III}$ of 0.43, 0.72, 0.45 g/cm$^3$, respectively, it is apparent that the $\rho_e^{I}$ and $\rho_e^{III}$ decrease with increasing particle size while $\rho_e^{II}$ is independent of particle size. The specific reasons are presented in a separate paper mainly dealing with the theoretical bases for three definitions of effective densities, which is not the focus of the present study.

5) *Line 239: By "different", do you mean "wider"?*

Response: Yes, it has been corrected accordingly. Please refer to Lines 262-265: "Besides, the result that the crystallized NaNO$_3$ particles are aspherical is supported by their FWHM values of the $D_{va}$ distributions which are consistent with that of aspherical (NH$_4$)$_2$SO$_4$ but wider than spherical PSL (Figures S1 and S2)."

6) *Lines 255-256: Add a reference to this methods sentence and move to the Methods section.*

Response: Thanks for the comment. Song et al. (1999) has been added accordingly.

7) *Line 281 and Table 1: Note that error should only be reported with 1 significant figure.*

Response: It has been changed accordingly. The values of effective density and size has been modified to 1 significant figure as suggested.

8) *Table 1 caption: State what the error corresponds to here.*

Response: Thanks for the comment. It has been added accordingly. The number after the sign of "±" is its standard deviation, which comes from the nine rounds of measurement. Please refer to Lines 447-449: "**Table 1.** $D_{ve}$ and its standard deviation for the eight particle types at $D_a$ values of 250.0 nm, 350.0 nm, 450.0 nm, and 550.0 nm from nine round measurement."

9) *Figure 1 caption: Add flow rates to the figure to make it more informative.*

Response: Thanks for the comment. It has been added accordingly.

10) *Figure S2b is missing.*

Response: Thanks for the comment. Sorry for this mistake. Fig. S2 do not include the Fig. S2b, so we delete the description of Fig. S2b in the caption. Please refer to Lines 27-28 in Supplement.

---

## Author Comment (AC1)

**Response to comments**

Response: We thank the reviewers for thoughtful suggestions and constructive criticism that have helped us improve our manuscript. Below we have detailed the responses and resulting edits to all of the reviews' comments. The review comments are listed in italics and black, followed by our responses in normal font and blue. To make it clear, the contents in revised manuscript are presented in quotes and normal font. Reference to line numbers are added to the revised manuscript.

**Referee#1:Johannes Schneider**

General Comments*: This technical note is appropriate for ACP. The combination of single particle results (via the clustering algorithms), effective density and shape factors is interesting and is worth to be exploited further. However, some important comments have to be addressed before this paper can be published:*

Specific comments:

*1) Both ZEL2006 and SCH2006 used the mobility diameter measured by a differential mobility analyzer (DMA) and not the AAC that measures the aerodynamic diameter, but the basic idea is the same. In SCH2006, we already presented the equation for rho_eff_II (your Equ. 8). Thus, please give proper credit to our work.*

Response: Thanks for your suggestion. We have included the citation (Schneider et al., 2006) as suggested. Please refer to Lines 82-83: "The detailed derivation of Eq. (8) was presented in Schneider et al. (2006)."

*2) Lines 68-76: You rely here on DeCarlo et al., 2004, but the numbering of the effective density is different. rho_e_I is the same, but you changed rho_e_II and rho_e_III compared to DeCarlo et al. Please use the same numbering to avoid confusion. Please also refer to Hand et al., 2002 who introduced your rho_e_III (which in de Carlo et al. is termed rho_eff_II).*

Response: Thanks for your suggestion. To avoid confusion, we change the numbering of the definitions of the effective density based on DeCarlo et al., 2004. Besides, we refer Hand and Kreidenweis (2002) to the definition of $\rho_e{}^{II}$. Please refer to Lines 68-76: "At present, three definitions of $\rho_e$ are introduced in atmospheric science (DeCarlo et al., 2004): the first definition ($\rho_e{}^{I}$) is the ratio of the measured particle mass ($m_p$) to the particle volume ($V$) calculated assuming a spherical particle with a diameter equal to the measured $D_m$; the second definition ($\rho_e{}^{II}$) is the ratio of $\rho$ to $\chi$ (Hand and Kreidenweis, 2002); and the third definition ($\rho_e{}^{III}$) is the ratio of $D_m$ and $D_{va}$, all of which are expressed in Eqs. (4)-(6), respectively. "

$$\rho_e^I = \frac{6m_p}{\pi D_m{}^3} \tag{4}$$

$$\rho_e^{II} = \frac{\rho_p}{\chi} \tag{5}$$

$$\rho_e^{III} = \frac{D_{va}}{D_m} \rho_0 \tag{6}$$

*3) Line 80, equation 8: rho_0 is wrong here, needs to be deleted to get the units right.*

Response: Thanks for your comment. It has been corrected accordingly.

*4) Line 81: "The detailed derivation will be presented in a separate paper". The derivation of Equ. 8 was given in deCarlo et al., 2004, and also in SCH2006, so please give proper reference here, and for completeness, give the derivation of Equ. 7 here as well.*

Response: Thanks for your comment. The citation and derivation of Eq. 7 have been added accordingly. Please refer to Lines 81-83: "The Eq. (7) is derived from combining the Eq. (1) with Eq. (4) in which $m_p$ is equal to $1/6\ \rho \cdot D_{ve}{}^3$. The detailed derivation of Eq. (8) was presented in Schneider et al. (2006). "

*5) Line 157, Eq. 11: To calculate D_ve, you need the Cunningham slip correction values here. How are they obtained? The differences between D_ve and D_a (e.g. Fig. S2) are rather large, so the Cunningham correction can not be neglected.*

Response: We agree with the comment. Actually, the Cunningham Slip Correction

Factor was not neglected in this study. The calculating process of the $D_{ve}$ is presented in the Section 2.3, please refer to Lines 169-182:

"Combining Eqs. (2) and (3), we obtain the following Eq. (10):

$$C_c(D_a)\frac{D_a^2}{D_{va}} = D_{ve}C_c(D_{ve})\frac{\chi_v}{\chi_t} \tag{10}$$

Based on the approximation between $\chi_v$ and $\chi_t$ ($\chi_v \approx \chi_t = \chi_a$) (DeCarlo et al., 2004), Eq. (10) becomes Eq. (11):

$$C_c(D_a)\frac{D_a^2}{D_{va}} = D_{ve}C_c(D_{ve}) \tag{11}$$

The Cunningham Slip Correction Factor is calculated by Eq. (12):

$$C_c(D) = 1 + \frac{\lambda}{D}\left(A + B \cdot \exp\left(\frac{C \cdot D}{\lambda}\right)\right), \tag{12}$$

where $\lambda$ is the mean free path of the gas molecules, and $A$, $B$ and $C$ are empirically determined constants specific to the analysis system. Substituting Eq. (12) into Eq. (11) obtains the Eq. (13).

$$\frac{D_a^2}{D_{va}} + \frac{D_a \cdot \lambda}{D_{va}}\left(A + B \cdot \exp\left(\frac{C \cdot D_a}{\lambda}\right)\right) = D_{ve} + \lambda\left(A + B \cdot \exp\left(\frac{C \cdot D_{ve}}{\lambda}\right)\right) \tag{13}$$

Thus if the $D_a$ and $D_{va}$ of an unknown particle are measured, its $D_{ve}$ could be calculated according to Eq. (13)."

*6) Line 161: I think that Equ. 12 results from combining Equ. 3 and Equ. 6. Correct?*

Response: Correct, and it has been clarified based on the above comments. Please refer to Lines 182-184: "Finally, the $\rho_e$ value of the particles is calculated by the $D_{va}$ and $D_{ve}$ values according to Eq. (14), which is obtained by combining Eq.(3) and Eq.(5): "

$$\rho_e = \frac{\rho_p}{\chi_a} = \frac{D_{va}}{\rho_0 \cdot D_{ve}} \tag{14}$$

*7) Line 180 – 182: Please clarify that for spherical particles like PSL, rho_e = rho (see deCarlo et al., Equ [43] and thereafter).*

Response: It has been clarified accordingly. Please refer to Lines 204-206: "The deviations of $\rho_{e,me}$ are determined to be 4.3%, -5.2%, -5.2%, and 4.3%, respectively, by comparing to the theoretical $\rho_e$ ($\rho_{e,th}$) that is equals to the $\rho_p$ for the spherical particles (i.e. 1.055 g/cm$^3$ of PSL particles)."

8) *Line 206-209: If different definitions of the effective densities are used, the statement "This pattern is divergent with the previous studies, which showed that effective density decreased as the size increasing" has to be removed or at least reworded.*

Response: Thanks for your comment. The statement has been reworded, please refer to Lines 235-237: "It is determined by the definition of effective density used in this study, which keeps constant as long as the $\chi_a$ does not change with particle size for pure compounds."

9) *Line 259-260: Please refer to the respective Figures in the Supporting Information.*

Response: It has been clarified accordingly. Please refer to Lines 282-284: "Details of the chemical composition and number fraction of the eight types of particles are presented in the Figures S3 and S4, respectively, which are discussed in the Supporting Information."

10) *Line 275-280: rho_e = rho_p/shape factor (Equ. 6). Thus, either the density of the particle material is a function of size, or the particle shape factor (or both). I think you can not rule out that the material density changes with size. SPMS is not quantitative, so particles of the same cluster type may have different quantitative composition (e.g. the ratio OC/EC or organic/inorganic). Thus, you can't tell whether the changing rho_ewith size is an effect of shape or composition*

Response: We agree with the comment. They have been revised to Lines 302-304: "Specifically, the $\rho_e$ of K-Na increases with $D_{ve}$, while the $\rho_e$ of OC-N-S and OC-EC-N-S decreases with $D_{ve}$, which may be influenced by the particle shape and/or the material density."

11) *Supplement: Fig.S2b) is missing.*

Response: Thanks for your comment. Sorry for this mistake. Fig. S2 do not include the Fig. S2b, so we delete the description of Fig.S2b in the caption. Please refer to Lines 27-28 in Supporting Information.

---

## Author Response (AR2)

**Response to comments**

Response: We thank the reviewer for his comments and recommendations to improve the manuscript. Below we have detailed the responses and resulting edits to all of the reviews' comments. The review comments are listed in italics and black, followed by our responses in normal font and blue. To make it clear, the contents in revised manuscript are presented in quotes and normal font.

**Referee#1: Johannes Schneider**

Specific comments: lines 169 - 185, the discussion of the Cunningham slip correction:please give a reference for equation (12). "A, B and C are empirically determined constants specific to the analysis system." Did you determine these constants for your system, and if so, how? What are the numbers for A, B, C that you have used?

Response: We have added the references and the values of A, B, C. Please refer to Lines 190-194:

Lines 190-194: "The Cunningham Slip Correction Factor is calculated by Eq. (12) (DeCarlo et al., 2004):

$$C_c(D) = 1 + \frac{\lambda}{D} \left( A + B \cdot \exp\left(\frac{C \cdot D}{\lambda}\right) \right), \tag{12}$$

where  $\lambda$  is the mean free path of the gas molecules, and *A*, *B* and *C* are empirically determined constants specific to the analysis system. The values of *A*, *B* and *C* are 2.33, 0.966, and -0.498, respectively, which are provided by the manual of the AAC."

**Reference**

DeCarlo, P. F., Slowik, J. G., Worsnop, D. R., Davidovits, P., and Jimenez, J. L.: Particle morphology and density characterization by combined mobility and aerodynamic diameter measurements. Part 1: Theory, Aerosol Sci. and Technol., 38, 1185-1205, https://doi.org/10.1080/027868290903907, 2004.

**Response to comments**

Response: We thank the reviewer for his/her thoughtful suggestions and constructive criticism that have helped us further improve our manuscript. Below we have detailed the responses and resulting edits to the review's comments. The comments are listed in italics and black, followed by our responses in normal font and blue. To make it clear, the contents in revised manuscript are presented in quotes and normal font.

**Referee#2: Anonymity**

General Comments: Peng et al present a revision of their manuscript combining an AAC and single-particle mass spectrometry to determine chemically-resolved single-particle effective densities. As noted previously by both reviewers, this work is interesting and useful. However, both reviewers pointed to a major manuscript weakness being limited citations to previous, highly relevant work and some explanations needing clarifications. In addition, several previous comments focused on stated uncertainties and errors in the results. Several of these issues have been corrected, but some still remain, as noted below.

1) The authors added sentences in the introduction discussing the use of particle light scattering to obtain effective density, and these added sentences on Lines 87-94 are useful. However, most previous and similar work has used a DMA with single-particle mass spectrometry, and both reviewers pointed to the need to discuss this previous work in the introduction to place the current similar work (that combines AAC with single-particle mass spectrometry) in context and make it clearer how this current work builds upon this previous work. However, the authors chose not to do this, with the exception of adding a statement that the derivation of Eq 8 was previously presented in Schneider et al (2006) (lines 82-83). I still believe it is a major weakness of the paper to not discuss previous work combining a DMA with single-particle mass spectrometry (see previous reviews) in this manuscript's introduction.

Response: Thanks for the comments. After careful consideration, we also think the

inclusion of previous work combining a DMA with single-particle mass spectrometry would be better for the intact, although it is different to our method. We have added the discussion of such work accordingly, please refer to Lines 83-94:

"A variety of methods are developed to characterize  $\rho_e^I$  and  $\rho_e^{III}$ , among which the more advanced methods are to achieve the measurement of the chemically-resolved effective density. Combining a single particle soot photometer (SP2) with a (volatility) tandem differential mobility analyser ((VT)DMA) can measure the  $\rho_e^I$  of particles mixed with soot (Zhang et al., 2016b; Wu et al., 2019; Han et al., 2019). The measurement of chemically-resolved  $\rho_e^{III}$  can be achieved by coupling a DMA with an on-line aerosol mass spectrometer including the single particle laser ablation time-offlight mass spectrometer (SPLAT I/II) (Zelenyuk et al., 2005; Zelenyuk et al., 2006; Alexander et al., 2016), aerosol mass spectrometer (AMS) (Dinar et al., 2006; Schneider et al., 2006; Kiselev et al., 2010), aerosol time-of-flight mass spectrometer (ATOFMS) (Spencer and Prather, 2006; Spencer et al., 2007), and single-particle aerosol mass spectrometry (SPAMS) (Zhang et al., 2016a; Zhai et al., 2017)."

2) Following on the previous comment, the authors did not rephrase the problematic statements that give the impression that this is the first work to measure chemically-resolved particle effective density (previous lines 90, 150-151, 298-299, now lines 98, ). These sentences include: "The aim of the present work is to develop a method to obtain Dve and pe." (line 98 in current manuscript) "These two properties cannot yet be measured for unknown particles by current techniques." (when referring to Dve and pe on lines 167-168 in the current manuscript) "...to first characterize the Dve, pe, and chemical compositions of atmospheric particles..." (lines 321-322 of the current manuscript) As previously requested, these misleading statements should be rephrased.

Response: Sorry for the misunderstanding. The problematic statements have been rephrased. Please refer to:

a) Line 109: "The aim of the present work is to develop a method to simultaneously obtain  $D_{ve}$  and  $\rho_e$ ."

b) Lines 183-184: "These two properties cannot yet be simultaneously measured for unknown particles by current techniques."

c) Lines 336-339: "Coupled with the ability of SPAMS to characterize the chemical composition of individual particles, we conducted a sample proof of the AAC-SPAMS equipment to first simultaneously characterize the  $D_{ve}$ ,  $\rho_e$  and chemical compositions of atmospheric particles, showing the potential application of this system in field observations."

3) Throughout the manuscript, error should be reported with one significant figure. In addition, while error was added to the PSL sizes on Lines 139-142, it is overstated. For example, 203.0 +/- 5.0 nm shows greater certainty than 203. +/- 5. nm, which would be in line with the manufacturer's stated uncertainty. Also, in the response, the authors state that they fixed error to be reported with one significant figure in Table 1, but all of the errors shown in this table still are 2-3 significant figures.

Response: Thanks for pointing out this. We have revised the description to "Dried spherical PSL (Nanosphere Size Standards, Duke Scientific Corp., Palo Alto) ( $\rho_p = 1.055 \text{ g/cm}^3$  and  $\chi = 1.0$ ) with  $D_{ve}$  values of  $203 \pm 5 \text{ nm}$ ,  $310 \pm 6 \text{ nm}$ ,  $510 \pm 5 \text{ nm}$ , and  $740 \pm 6 \text{ nm}$  were used in the AAC-SPAMS system, and the  $D_{ve}$  was verified by Scanning Mobility Particles Sizer (Model 3938, TSI Inc., USA).", which is in line with the manufacturer's stated uncertainty.

For the data shown in Table 1, as shown in the previous response, we indicated that the resolution for the measured  $D_{ve}$  values can be as small as < 1 nm, and thus we have revised the mean values and standard deviation calculated from several measurements to 1 decimal place in the revised manuscript. Note that the decimal place for the mean  $D_{ve}$  is consistent with the measurement error in the AAC-SPAMS system.

4) The authors state in their response that "Considering the precision of the PSL size is less than that of the instrument of AAC, the discrepancy between the measured value (from AAC-SPAMS) and the true value (the size and density of PSL) is used to represent the measurement uncertainty, which is presented in Figure 2." However, the error in the AAC size needs to be stated in (added to) to the manuscript for the reader to evaluate this statement. It appears that these measurement uncertainties were not propagated through to the reported deviations in size and density in Figure 2, so the reader cannot evaluate where the major source of error is originating, which is important for the results. The authors at least need to state the errors in the manuscript to justify their method. It is also critical that these uncertainties be clearly explained so that the number of decimal places used in presenting the data in the results can be properly evaluated.

Response: We agree with the comment that it is critical to evaluate the measurement error accurately. Firstly, we have added the measurement uncertainty of the AAC. Please refer to Lines 157-161:

"Besides, to present the measurement uncertainty of the AAC, the  $D_a$  values of these PSL particles were measured to be  $212.8 \pm 0.2$ ,  $324.7 \pm 0.4$ ,  $529.9 \pm 0.4$ , and  $767.5 \pm 0.4$ , respectively, by the system of AAC- Condensation Particle Counter (CPC), which shows that the AAC has the deviations of 1.1%, 1.3%, 0.8%, and 0.7% for determining the  $D_a$  values of the particles.".

Then, the uncertainties for the Gaussian fitting to obtain the peak  $D_{va}$  for PSL was estimated to be of 6.6%, 4.4%, 2.3% and 2.2%.

Finally, we calculated the deviations between the theoretical  $D_{ve}$  ( $D_{ve,th}$ ) and  $D_{ve,me}$ , which is < 1% and <6%, respectively. And thus we may conclude that the errors from AAC (~1%) and fitting of  $D_a$  (~2-7%) should explain the errors of  $D_{ve,me}$  (< 1%) and  $\rho_{e,me}$  (<6%) measured by the AAC-SPAMS system, as discussed in section 3.1.

5) The ambient sampling section of the Methods (Sec 2.3) needs to state the dates of sampling, as previously requested, since the seasonality will impact particle composition, for example.

Response: Thanks for the comment. We have added the dates of sampling. Please refer

to Lines 172-173:

"The sampling time for the particles of each  $D_a$  was approximately 10 minutes. From July 6th to 8th, 2019, approximately 129,869 ionized particles were obtained from nine rounds of measurement."

6) Lines 304-306: Errors were reported for all particle types in the previous version of the manuscript, but now these are missing for the Amine, EC-N-S, and EC-S particle types for an unknown reason.

Response: Thanks for pointing out this mistake. They have been added. Please refer to Lines 320-325:

"Additionally, the average  $\rho_e$  of each type of particle is in the order from small to large:  $1.2 \pm 0.2$  g/cm3 for OC-EC-N-S,  $1.3 \pm 0.2$  g/cm3 for OC-N-S,  $1.4 \pm 0.1$  g/cm3 for K-rich,  $1.4 \pm 0.1$  g/cm3 for Amine,  $1.5 \pm 0.1$  g/cm3 for EC-N-S,  $1.5 \pm 0.1$  g/cm3 for EC-S,  $1.6 \pm 0.1$  g/cm3 for K-Na and  $1.6 \pm 0.1$  g/cm3 for Metal-rich. It is reasonable to find that the average  $\rho_e$  of internally mixed particles distributes in the range of their material densities ( $\rho_m$ )."

7) Consistency between Methods and Figure 1 caption: It is now stated that the system flow rate is 0.3 lpm, but line 153 states an "overall sampling flow of 3 lpm". Line 124 states "The exhaust flow from the AAC was about 0.3 lpm." However, the sampling flow rates of the AAC and SPAMS are not stated in Section 2.1. Please add these to Section 2.1, clarify the flow rates at each part of the system in Figure 1, and clarify the discrepancy between the figure caption and line 153.

Response: Thanks for pointing out this, which is due to our incorrect description in the *Methods*. As stated in the figure caption, the sampling flow in the system of AAC-SPAMS is 0.3 lpm, which mean that the sampling flow from AAC to SPAMS was 0.3 lpm. The overall sampling flow of 3 lpm refers to the sampling flow for the  $PM_{2.5}$  cyclone inlet. To reduce the residence time of particles in the conductive silicone, a

diaphragm pump was connected in parallel with the diffusion drying tube and ran at a flow rate of 2.7 lpm, which was not present in Figure 1. We have revised the description to "The sampling flow from the  $PM_{2.5}$  cyclone inlet was 3 lpm, and the residence time in the conductive silicone tube was approximately 5 seconds.", please refer to Lines 168-170.